# Multicentre, longitudinal, observational cohort study to examine the relationship between neutrophil function and sepsis in adults and children with severe thermal injuries: a protocol for the Scientific Investigation of the Biological Pathways Following Thermal Injury-2 (SIFTI-2) study

Jon Hazeldine ![ORCID],[1,2] Kirsty C McGee,[2] Khaled Al-Tarrah,[2] Tarek Hassouna,[3] Krupali Patel,[3] Rizwana Imran ![ORCID],[3] Jonathan R B Bishop,[1] Amy Bamford,[1,3] David Barnes,[4] Yvonne Wilson,[5] Paul Harrison,[2,6] Janet M Lord ![ORCID],[1,2,6] Naiem S Moiemen[1,3,6]

**Correspondence to**
Professor Naiem S Moiemen;
naiem.moiemen2@nhs.net

## ABSTRACT

**Introduction** Burn-induced changes in the phenotype and function of neutrophils, cells which provide front-line protection against rapidly dividing bacterial infections, are emerging as potential biomarkers for the early prediction of sepsis. In a longitudinal study of adult burns patients, we recently demonstrated that a combined measurement of neutrophil phagocytic capacity, immature granulocyte (IG) count and plasma cell-free DNA (cfDNA) levels on the day of injury gave good discriminatory power for the prediction of later sepsis development. However, limited by a small sample size, single-centre design and focus on adult burns patients, these biomarkers require prospective validation in a larger patient cohort. The Scientific Investigation of the Biological Pathways Following Thermal Injury-2 study aims to prospectively validate neutrophil phagocytic activity, IG count and plasma cfDNA levels as early prognostic biomarkers of sepsis in thermally injured adult and paediatric patients.

**Methods and analysis** This multicentre, longitudinal, observational cohort study will enrol 245 paediatric and adult patients with moderate to severe burns within 24 hours of injury. Blood samples will be obtained at 19 postinjury time points (days 1–14, day 28, months 3, 6, 12 and 24) and analysed for neutrophil phagocytic activity, IG count and cfDNA levels. Patients will be screened daily for sepsis using the 2007 American Burn Association diagnostic criteria for sepsis. In addition, daily multiple organ dysfunction syndrome and Sequential Organ Failure Assessment Scores will be recorded relationships between neutrophil phagocytic activity, IG count and plasma cfDNA levels on day 1 of injury and the development of sepsis will be examined using logistic regression models.

### Strengths and limitations of this study

► This study's main strength is its immediate (days 1–28) and long-term (months 3–24) analysis of the immune and inflammatory response to severe thermal injury in both paediatric and adult patients.

► Analysis of immediate and long-term postinjury blood samples will allow for kinetic profiling of the immune and inflammatory response to severe thermal injury.

► Centred on the analysis of innate immune function, this study will offer little insight into the immediate and long-term changes in the adaptive immune system following thermal injury.

**Ethics and dissemination** This study received ethics approval from the West Midlands, Coventry and Warwickshire Research Ethics Committee (REC reference:16/WM/0217). Findings will be presented at national and international conferences, and submitted for publication in peer-reviewed journals.
**Trial registration number** NCT04693442.

## INTRODUCTION

Built on such clinical practices as early burn excision and wound closure, goal-directed fluid resuscitation and early enteral feeding, modern day burn care has markedly improved the initial outcomes of thermally injured patients, who now survive injuries that were fatal less than 30 years ago.[1] However,

with such unprecedented survival rates, new challenges in the management of hospitalised burns patients have emerged. With an incidence rate of between 23%–63% in paediatric burns patients[2–4] and 8%–42.5% in adult burns patients,[5] sepsis is a common complication following thermal injury. Defined as 'life-threatening organ dysfunction caused by a dysregulated host response to infection',[6] postburn sepsis is associated with significantly increased lengths of intensive care unit and hospital stay, and is the leading cause of mortality among hospitalised burns patients, accounting for between 28%–65% of deaths in adult patients[5] and 47% of deaths in paediatric patients.[7] Understanding the mechanisms underlying the development and progression of sepsis is therefore critical if we are to improve patient outcomes postburn.

Characterised by elevated circulating concentrations of proinflammatory cytokines and immune activation, severe thermal injury results in a systemic inflammatory response syndrome (SIRS).[8] This immediate and persistent inflammatory and hypermetabolic state triggers a physiological response in burns patients that mirrors that observed in patients with sepsis. Indeed, fever, tachycardia and tachypnoea, which were included by the American Burn Association (ABA) in their latest criteria for diagnosing sepsis among hospitalised burns patients,[9] are examples of some of the clinical manifestations used to identify patients with SIRS.[10] Thus, in response to a sterile insult, burns patients exhibit many of the classic diagnostic biomarkers of sepsis, thereby making the diagnosis of this secondary complication extremely difficult. Faced with this clinical challenge, a number of studies have searched for prognostic and diagnostic biomarkers of postburn sepsis. To date, a number of potential candidates, which include procalcitonin, tumour necrosis factor-alpha, interleukin (IL)-6 and IL-8, have been identified.[11] However, as with clinical parameters, these markers of inflammation are elevated during the initial SIRS response triggered by sterile burn injury,[8] which may explain why statistical models built on these biomarkers to predict postburn sepsis in children and adults exhibit poor sensitivity and specificity.[12 13] Given that for every hour delay in the treatment of sepsis, the risk of death increases by 4%,[14] there is an urgent need to identify novel biomarkers that can be used to accurately predict sepsis in thermally injured patients.

Providing immediate front-line protection against rapidly dividing bacterial and fungal infections, neutrophils are critical effector cells of the innate immune system. Severe thermal injury has been shown to lead to impairments in a range of neutrophil functions such as chemotaxis,[15–17] phagocytosis,[17–19] generation of reactive oxygen species[18 19] and the formation of neutrophil extracellular traps, which contribute to circulating cell-free DNA (cfDNA).[18] Given their critical role in the elimination of bacterial infections, this postburn reduction in neutrophil function has been proposed to underlie the increased susceptibility of thermally injured patients to nosocomial infection and sepsis. Interestingly, results

from a number of recent longitudinal studies suggest that an early assessment of the phenotype and/or function of the circulating neutrophil pool can be used to predict, diagnose and/or monitor postburn sepsis.[15 18 20] In a cohort of 13 patients with major burns, Jones *et al* demonstrated that neutrophils isolated from patients during a septic episode displayed a spontaneous migratory phenotype that was significantly different to that recorded for neutrophils isolated from burns patients with or without SIRS.[15] Importantly, demonstrating both its prognostic utility and monitoring potential, this phenotype was also observed in patients prior to sepsis diagnosis and was corrected following effective antibiotic treatment.[15] Extending these observations, we recently demonstrated, in a cohort of adult burns patients, that a combined measure of neutrophil phagocytic capacity, immature granulocyte (IG) count and plasma cfDNA levels exhibited strong discriminatory power (area under receiver operating characteristic, AUROC 0.935) on day 1 of injury for distinguishing between patients who did or did not develop sepsis.[18] Moreover, we found that a statistical model that combined the variables of neutrophil phagocytosis and IG count with a clinical scoring metric (revised Baux score) provided discriminatory power (AUROC 0.986) that was greater than any variable alone.[18] Although highlighting the potential utility of combined clinical and immune biomarker data for the early prediction of sepsis, our original study was designed to be exploratory and hypothesis generating in nature rather than confirmatory, and as such lacked a formal power calculation. Moreover, the study was limited by its small sample size, single-centre design and the inclusion of data only from adult burns patients. Thus, a much larger study that prospectively validates this novel suite of biomarkers in both adult and paediatric burns patients is required.

### Primary objective

To validate neutrophil phagocytic activity, IG count and plasma cfDNA levels as early biomarkers of sepsis in adult and paediatric thermally injured patients.

## METHODS AND ANALYSIS
### Study design

The Scientific Investigation of the Biological Pathways Following Thermal Injury-2 (SIFTI-2) study is a multicentre, prospective, longitudinal observational cohort study of children and adult patients with moderate and severe burn injury: ≥15% of the total body surface area (TBSA) in adults and≥20% TBSA in children.

### Study population

Study participants are paediatric (1–15 years) and adult (≥16 years) burns patients presenting within 24 hours of injury to participating burns centres that include but are not limited to: (1) West Midlands Regional Burns Centre, Queen Elizabeth Hospital Birmingham

(QEHB), Birmingham, UK; (2) Paediatric Burns Centre, Birmingham Children's Hospital (BCH), Birmingham; (3) St Andrew's Centre for Plastic Surgery and Burns, Broomfield Hospital, Chelmsford, Essex, UK.

Screening of the enrolment log generated for patients recruited into our previous observational study,[18] revealed a predominantly male (61%) and Caucasian (79%) patient cohort. Given that the West Midlands Regional Burns Centre will again serve as the primary site of patient recruitment, we anticipate a similar patient demographic for the SIFTI-2 study.

Patients will be identified by research clinicians or the admitting doctor from the burns research team and screened for study eligibility using the following inclusion and exclusion criteria:

### Inclusion
► Patients aged 1–15 years admitted with a ≥20% TBSA burn.
► Patients aged 16 years and over admitted with a ≥15% TBSA burn.

### Exclusion
► Associated multiple injuries with an injury severity score (ISS) >25.
► Decision not to treat due to the severity of the injury. This decision will be made at the time of hospital admission.
► Patients with chemical or deep electrical burns.
► Patients receiving glucocorticoid treatment.
► Patients with active malignancy.
► Patients with multiple limb amputations (amputations would skew the calculation of TBSA scores).
► Patients with known long-term infections (eg, hepatitis B and C, HIV).

Alongside burns patients, a maximum of 100 adults and 10 children will be recruited to serve as a cohort of age-matched healthy controls (HCs), with the only exclusion criterion being current treatment with anti-coagulant medication or an acute infection.

Adult HCs can be recruited from age 16 years, with no upper age limit. Enrolment of up to 100 healthy adults will enable us to undertake diverse sampling of the adult population, allowing us to control for as many of the variations that encompass this broad age range as possible (eg, general physical condition, concurrent medical conditions and drug therapies, past exposure to disease or injury and different degrees of physiological ageing). Adult HC volunteers will be recruited through advertisements disseminated at the QEHB, BCH, the University of Birmingham campus and via online newsletters. When an expression of interest for study participation is received by the burns research team, a patient information leaflet (PIL) detailing the conditions of the study will be sent to the volunteer. All subjects who receive a PIL will be recorded on a screening log that will be held in the site file at the QEHB. Adult volunteers will attend the QEHB to participate in the study.

The paediatric HC group will consist of children undergoing elective plastic surgery procedures at the BCH. These surgeries will include mole removal, correction of prominent ears, revision of plastic surgery and cosmetic type surgery. In terms of age range and a lack of other health conditions or ongoing medical treatments, paediatric HCs will represent a more homogeneous cohort when compared with our adult HC group. Thus, we expect to observe much less variability in the baseline immune profiles of paediatric HCs. This factor, combined with the complexity of recruiting paediatric HCs, is the reasoning behind our decision to enrol a maximum of 10 healthy paediatric volunteers into the SIFTI-2 study.

## Consent
Patients who meet study inclusion criteria and are deemed suitable for enrolment into the SIFTI-2 study will be assessed for capacity and assigned to one of the following three groups for the appropriate informed consent process: (1) adult patients with mental capacity, (2) adult patients lacking mental capacity or (3) children (aged<16 years). Details on the informed consent procedure for each of these groups are outlined below and summarised in figure 1.

### Adult patients with mental capacity
Patients will be provided with a PIL that outlines the research study. If, after a discussion with the burns research team, the patient agrees to participate, they will be provided with and asked to sign a study consent form.

### Adult patients lacking mental capacity
On hospital admission, the patient's capacity will be assessed in accordance with the Mental Health Capacity Act (2005). Given the nature of this study, patients, either on, or shortly after their arrival at the emergency department, may lack capacity due to the severity of their injuries or as a consequence of sedation and/or ventilation. A patient may also lack capacity due to a pre-existing comorbidity. If a patient does not have the capacity to make an informed decision, the burns research team will approach a patient's personal consultee (eg, next of kin, relative, carer or friend). In instances where a personal consultee is not available, a nominated consultee will be sought. Nominated consultees are considered to be medical professionals that have no connection to the research study, but have an understanding of its implications on the participant. Examples of nominated consultees who will be approached include: emergency department doctors, intensive care doctors or doctors from the burns team that are not directly involved in the SIFTI-2 study. The rationale for the use of nominated consultees is that the physiological response to burn trauma is ongoing from the time of injury. As such, while efforts are being made to contact their next of kin, this system of consent enables research blood samples to be obtained during the early postinjury phase that would otherwise be missed.

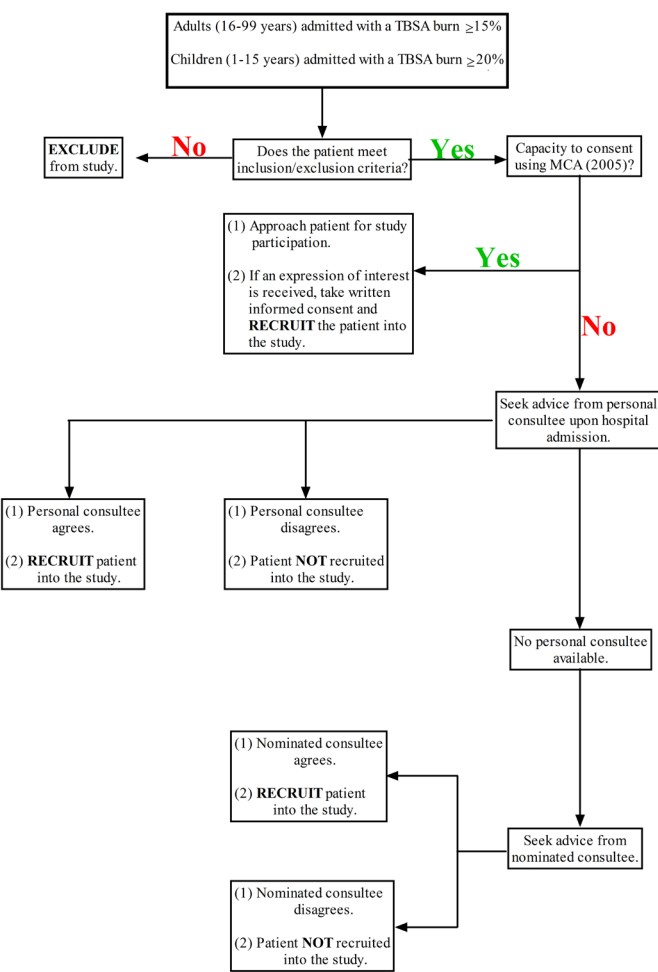

**Figure 1** Flow chart of patient screening and the consent procedure. MCA, Mental Capacity Act; TBSA, total body surface area.

Once a personal or nominated consultee has been identified, they will be provided with a study specific information leaflet. If in their opinion, the patient would have no objection to being recruited into a research study, the consultee will be asked to sign a declaration form. When the patient regains capacity, the study will be explained to them and they will be asked for their consent for the data obtained so far to be used and for any future sampling. If the participant declines consent, the data collected will be deleted and no further sampling undertaken. In instances where the patient does not regain capacity, their data will be included into the study in accordance with the legal consultee's assent.

### Children (aged <16 years)
A child will be recruited into the SIFTI-2 study in one of three ways:

#### Parental/legal guardian consent
All children will be consented into the SIFTI-2 study by their parent or legal guardian. If, after an explanation of the study, the parents/guardians have no objection to their child's involvement in the study, they will be asked to sign the parental consent form on behalf of the child.

#### Child consent
As all children will be consented into the study by their parent or legal guardian, no child can self-consent for study enrolment. However, children deemed psychologically and emotionally mature, and who meet the Gillick Competence criteria,[21] will be able to consent to their study involvement. The child will be provided with information on the study and, if after a discussion with the burns research team, agree to participate, they will be asked to sign the child consent form. This is to ensure that children of a certain level of maturity understand what is being proposed and that they agree with joining the study. Thus, the child's consent is a matter of confirming their agreement with the adult consent. If any child, who was deemed Gillick competent, does not wish to give their consent, then they would not be recruited into the study, even if the legally responsible adult had consented. If the child is not consenting, they do not join the study.

#### Deferred consent
A model of deferred consent will be used in situations where the clinical or research team feel initial parental/guardian consent is not appropriate and may cause further undue stress. Under this model, children will be enrolled by the burns or critical care team on admission so that day 1 and day 2 blood samples can be obtained. Parents/guardians will then be approached about the study and provided with an information sheet within 48 hours of their child's hospital admission. Parental/guardian consent should be obtained within this 48-hour period. If consent for future study involvement is refused, then consent to keep the initial clinical and laboratory data will be sought. If refused, then all data will be destroyed.

### Withdrawal of consent
Participants may withdraw or be withdrawn from the study at any time, the reason for which will be collected and recorded in the electronic case report form (CRF). Any data or samples collected at the time of withdrawal may still be included in the data analysis, unless a participant specifically withdraws consent. Participants will be asked to clarify this at the point of withdrawal.

It is possible that a participant maybe withdrawn from the study by the Sponsor, for non-compliance with study procedure. Such criteria for this include: (1) subject withdrawal of consent, (2) subject non-compliance, (3) any medical condition not compatible with continuation of the study and (4) as a result of safety review and recommendation.

### Sample size
The sample size for this study is based on data collected from adult patients recruited into the SIFTI-1 study.[18] Of the 57 adult burns patients who survived for at least 7 days following study enrolment, 35 developed sepsis, equating to a prevalence of 61%. Of these, 33 patients, with a TBSA burn ≥15%, had measurements of neutrophil phagocytic capacity, plasma cfDNA levels and IG count recorded on

day 1 of injury. These data provided good discriminatory power to identify septic patients, with logistic regression modelling providing an area under the ROC of 0.935. Adjusted for shrinkage following internal bootstrap model validation to account for potential overfitting, this model exhibited a specificity of 0.75 and a sensitivity of 1.00. Based on this information, we computed the sample sizes required to estimate a 95% CI for sensitivity (with a maximum width of 10%) and for specificity (with a maximum width of 20%) for prevalence rates of sepsis ranging from 50% to 70%. Assuming a prevalence of sepsis of at least 60% and that the 'true' sensitivity is at least 90% and the 'true' specificity is at least 70%, these analyses revealed a total of 220 patients would allow us to reliably rule out a sensitivity less than 84% and a specificity of less than 59%. Assuming that the 10% drop-out rate we recorded for patients over the first 7 days of the SIFTI-1 study holds true for this follow-on study, then a total sample size of 245 burns patients, which includes both adult and paediatric age groups, is required for the SIFTI-2 study.

### Data collection and patient outcomes

In this prospective longitudinal study, data will be acquired from burns patients at the following 19 postinjury time points: days 1–14, day 28 (±3 days) and months 3, 6, 12 and 24 (±2 weeks). Patient enrolment into the study began on 27 November 2016, with an estimated end date of December 2025.

Across the settings of paediatric and adult burns, a significant number of studies have undertaken serial blood sampling to investigate the short (<1–7 days), medium (8–34 days) and long (35–1100 days)-term inflammatory response to thermal injury and its relationship with patient outcome.[22–28] In contrast, very little data currently exist on the acute and long-term effects of thermal injury on the immune system.[19 24 29] Thus, generating data on the kinetics of the immune response to burn injury via serial sampling in both our adult and paediatric cohorts will allow us to not only establish how long burn-induced changes in immunity persist for, when compared with HCs, but also compare the immune response of septic and non-septic burns patients over time. Generating and interrogating such novel data may provide mechanistic insights into the long-term health complications reported by survivors of burn trauma, which include an increased susceptibility to respiratory infections.[30]

### Demographic and physiological data

Within 24 hours of injury, information on patient demographics, injury characteristics and baseline physiological data will be collected and recorded in paper and electronic CRFs. At subsequent study time points, physiological data will be collected and recorded in the electronic CRF. Table 1 summarises the data that will be recorded for all patients.

For adult HCs, the following clinical information will be recorded: date of birth, gender, weight in kilograms, height in metres and smoking status (active, ex-smoker or never smoked). The same details in addition to information relating to past medical history/congenital anomaly history will be recorded for paediatric HCs.

### Blood sampling

The schedule for blood sampling of adult and paediatric burns patients and the blood volumes that will

**Table 1** Demographic and clinical data that will be recorded in patient case report forms following enrolment into the SIFTI-2 study

| Day 1 (within 24 hours of injury) | | | Days 2–14 | Day 28, months 3, 6, 12, 24 |
|---|---|---|---|---|
| **Demographic data** | **Injury characteristics** | **Physiological data** | **Physiological data** | **Physiological data** |
| Age<br>Sex<br>Ethnicity<br>Time from injury to arrival at burn centre<br>Medical comorbidities<br>History of alcohol or substance misuse | % TBSA<br>% Body surface area full thickness burn<br>Burn mechanism<br>Presence of inhalation injury and severity<br>ABSI<br>APACHE II<br>AIS<br>ISS<br>NISS<br>PIMS-2<br>Revised Baux score<br>GCS score | Heart rate (bpm)<br>Blood pressure (mm Hg)<br>Mean arterial pressure (mm Hg)<br>Oxygen saturation (%)<br>Respiratory rate (breaths/minute)<br>Temperature (°C)<br>Blood glucose (mMol/L)<br>Height (m)<br>Weight (kg)<br>BMI (kg/m$^2$) | Heart rate (bpm)<br>Blood pressure (mm Hg)<br>Mean arterial pressure (mm Hg)<br>Oxygen saturation (%)<br>Respiratory rate (breaths/minute)<br>Temperature (°C)<br>Blood glucose (mMol/L)<br>Height (m)<br>Weight (kg)<br>BMI (kg/m$^2$)<br>Mode of ventilation<br>Early Warning Score<br>Use of haemofiltration or dialysis<br>Date of discharge/Date of death | Heart rate (bpm)<br>Blood pressure (mm Hg)<br>Mean arterial pressure (mm Hg)<br>Oxygen saturation (%)<br>Respiratory rate (breaths/minute)<br>Temperature (°C)<br>Blood glucose (mMol/L)<br>Height (m)<br>Weight (kg)<br>BMI (kg/m$^2$) |

PIMS-2 scores will be calculated for children<16 years. % TBSA will be calculated using the Lund and Browder chart.
ABSI, Abbreviated Burn Severity Index; AIS, Abbreviated Injury Scale; BMI, body mass index; GCS, Glasgow Coma Scale; APACHE II, Acute Physiology And Chronic Health Evaluation II; ISS, Injury Severity Score; NISS, New Injury Severity Score; PIMS-2, Paediatric Index of Mortality Score-2; SIFTI-2, Scientific Investigation of the Biological Pathways Following Thermal Injury-2; TBSA, total body surface area.

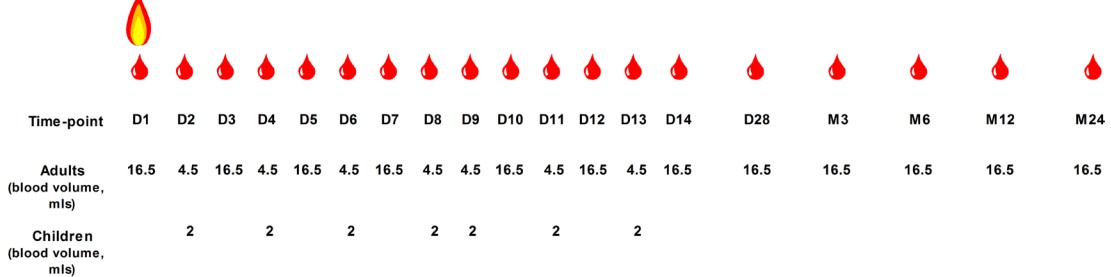

**Figure 2** Blood sampling schedule for paediatric and adult burns patients. For children, the volume of blood that will be acquired on days 1, 3, 5, 7, 10, 12, 14 and 28 as well as months 3, 6, 12 and 24 postburn will be dependent on their weight.

be obtained are outlined in figure 2. For children, the volume of blood that will be acquired on days 1, 3, 5, 7, 10, 12, 14 and 28 as well as months 3, 6, 12 and 24 post-burn will be dependent on their weight (table 2). For adult burns patients treated with low molecular weight heparin, blood samples on study days 5, 10 and 14 will be obtained 3 hours post-treatment. Blood samples will be taken following local standard operating procedures either from an arterial or central venous line or via venepuncture. The source of the blood samples will be dependent on the type of intravenous access the patient has. Posthospital discharge, research follow-up visits will take place during the patient's clinical follow-up appointments where possible. For HCs, 9 and 18 mL blood samples will be acquired from paediatric and adult volunteers, respectively, by venepuncture.

### Laboratory tests for the assessment of IG count, neutrophil phagocytic activity and cfDNA levels

Patient samples will be analysed at the University of Birmingham laboratories based at the QEHB. Samples will be accompanied by a pseudonymous form with no patient identifiable data. IG counts and neutrophil phagocytic activity will be assessed on the day of patient sampling, with blood samples processed immediately on their arrival into the laboratory. Assays for the measurement of cfDNA levels will be performed in two batches, details of which are outlined below.

### IG counts

Whole blood cell counts will be performed on citrated whole blood using a Sysmex XN-1000 haematology analyser (Sysmex UK, Milton Keynes, UK), which defines IGs as promyelocytes, myelocytes and metamyelocytes within the white cell differential channel. Instrument performance will be ensured by daily measurement of quality control material (XN Check) and via participation in an external quality assurance scheme (UKNEQAS, Watford, UK).

### Neutrophil phagocytic activity

Neutrophil phagocytosis of fluorescently labelled opsonised *Escherichia coli* will be measured on heparinised whole blood using the commercially available PhagoTEST assay (BD Biosciences, Oxford, UK). To account for postburn leucocytosis, patient's white cell counts will be normalised to those of HCs in order to ensure consistent leucocyte to bacteria ratios across all assays. By 'quenching' the fluorescent signal emitted from bacteria bound to the external plasma membrane of neutrophils, inclusion of a trypan blue staining step within this assay ensures that the fluorescent signal measured by flow cytometry is derived only from phagocytosed bacteria.

For the acquisition of raw data, a single flow cytometry template with specified gating plots and event limits will be used by all researchers analysing patient and HC blood samples. Gated based on their distinctive sideward scatter/forward scatter properties, a total of 10 000 neutrophils will be analysed and data relating to both the percentage of cells that have performed phagocytosis (% phagocytosis) and their mean fluorescence intensity (an indicator of the number of ingested bacteria per neutrophil) will be recorded. This information will then be used

**Table 2** Blood volumes to be obtained from thermally injured children (<16 years) enrolled into the SIFTI-2 study

| Age range | Weight range | Maximum blood volume that can be taken over 30 days* | Days 2, 4, 6, 8, 9, 11, 13 Blood volume to be taken | Days 1, 3, 5, 7, 10, 12, 14, 28, months 3, 6, 12, 24 Blood volume to be taken |
|---|---|---|---|---|
| 1–5 years | 10–18 kg | 40–72 mL | 2 mL | 3.25 mL |
| 5–7 years | 19–29 kg | 76–116 mL | 2 mL | 7.75 mL |
| 8–12 years | 30–40 kg | 120–160 mL | 2 mL | 13.25 mL |
| 12–15 years | >40 kg | >160 mL | 2 mL | 18.25 mL |

*The maximum blood volume that can be taken for research over a 30-day period is 5% of a patients total blood volume (TBV). The TBV of a child is around 75–80 mL/kg.
SIFTI-2, Scientific Investigation of the Biological Pathways Following Thermal Injury-2.

to calculate neutrophil phagocytic capacity using the following equation: (% phagocytosis/100) × mean fluorescence intensity.

Prior to flow cytometric analysis of all patient and HC blood samples, BD CS&T beads (BD Biosciences) will be run on the flow cytometer. These beads will ensure a standardised method of quality control will be performed on our instrument in respect to its optics, electronics and fluidics.

### CfDNA measurements

The concentration of total cfDNA), mitochondrial-derived DNA (mtDNA) and nuclear-derived DNA (nDNA) will be measured in platelet-free plasma (PFP) prepared from citrate anticoagulated blood samples. PFP will be stored at −80°C prior to analysis, which will be performed in two phases (phase 1: analysis of samples collected 1–28 days postburn; phase 2: analysis of samples collected at months 3–24 postburn).

Total cfDNA concentrations will be determined in duplicate PFP samples using an in-house SYTOX Green dye based fluorometric assay as described previously.[18] Included within each assay will be: (1) a λ-DNA standard curve (0–1000 ng/mL) for calibration, (2) PFP samples from HCs, stored under the same conditions as our patient samples, to serve as reference values and (3) a 'buffer only' negative control. The mean fluorescence value derived from the negative control will be subtracted from all HC and patient data prior to the calculation of DNA concentration. A selection of patient and HC PFP samples will be analysed across assays to allow for the calculation of interassay and intra-assay coefficients of variation, which currently stand at 5.3% and 5.1%, respectively. Analysis of 32 PFP samples using our in-house assay quantified cfDNA to concentrations that were comparable to those measured using a CE-marked commercially available assay (Trillium Diagnostics; Spearman's R=0.78, p<0.0001).

Using primer sets specific for the genes encoding cytochrome b and β-globin, real-time PCR will be performed to determine the concentration of mtDNA and nDNA, respectively.[18] Included within each assay will be: (1) standard curves generated using DNA isolated from purified mitochondria and nuclei to enable quantification of mtDNA and nDNA concentrations, (2) DNA isolated from HC PFP samples, stored under the same conditions as our patient samples, to serve as reference values and (3) a 'buffer only' negative control. PFP samples from a selection of HCs and patients will be analysed across assays to allow for the calculation of interassay and intra-assay coefficients of variation. Our cytochrome b and β-globin primer sequences have no significant homology with DNA found in any bacterial species according to information provided by the Basic Local Alignment Search Tool.

### Outcomes

In line with our original study,[18] sepsis will be defined according to the 2007 ABA diagnostic criteria for sepsis in burns.[9] A diagnosis of sepsis will be made when >3 of the following criteria are met along with a positive bacterial culture or when a clinical response to antibiotics is detected:

► Temperature (>39°C or<36°C).
► Progressive tachycardia (adults, >110 beats per minute; children, >2 SD above age-specific norms (85% of age-adjusted maximum heart rate)).
► Progressive tachypnoea (adults, >25 breaths per minute not ventilated or minute ventilation >12 L/ minute for ventilated patients; children, 2 SD above age specific norms (85% of age-adjusted maximum respiratory rate)).
► Thrombocytopenia (Adults, platelet count <150x10$^9$/L (this will not be applied until 3 days after initial resuscitation; children, <2 SD below age-specific norms).
► Hyperglycaemia (untreated plasma glucose >200 mg/ dL or insulin resistance as defined by either (1) >7 units of intravenous insulin per hour or (2) significant resistance to insulin (>25% increase in insulin requirements over 24 hours).
► Feed intolerance (inability to continue enteral feeding >24 hours defined using (1) abdominal distension, (2) enteral feeding intolerance (adults, two times the feeding rate; children, residual >150 mL/hour) or (3) uncontrollable diarrhoea (>2500 mL/day for adults or >400 mL/day in children).

Blood cultures will be taken whenever the temperature of a patient is >38.5°C and they will be acquired from a clean venepuncture site where possible using a sterile technique. If such a site is not accessible, due to the patients injuries, then blood cultures will be obtained from a pre-existing central venous catheter. The results will be interpreted following detailed discussions with the microbiology team.

Burn wound infection/sepsis will be documented in the CRF after assessment by a senior surgeon together with information attained from the relevant microbiology wound swabs/tissue biopsies. In addition, time to 95% wound healing will be documented as an overall measure of the patients care and for wound related complications such as wound sepsis.

Alongside recording data that will allow for the diagnosis of sepsis using the 2007 ABA consensus, clinical data will be collected from all patients so that the following clinical scores can be calculated: (1) Early Warning Score, (2) multiple organ dysfunction syndrome, (3) Sequential Organ Failure Assessment score, (4) Marshall Score and (5) SIRS Score. Comprehensive daily records of different scores will allow for more detailed analyses should a sepsis score for burns patients be validated in the future. This is particularly important for children, given that applying the ABA consensus score is challenging in this patient population, where organ dysfunction may be more indicative of sepsis.[31]

## Data analysis

Data will be assessed for normality using the Shapiro-Wilk test. Normally distributed data will be presented as mean values with SD. Non-normally distributed data will be presented as median values with IQR. Comparisons of baseline continuous data (eg, demographics, injury characteristics) between two groups (eg, HCs vs burns patients or septic vs non-septic patients) will be conducted using an unpaired Student's t-test for normally distributed data or the Mann-Whitney U test for non-normally distributed data. Categorical baseline data will be compared using a $\chi^2$ test. Laboratory data obtained from HCs and patients across the different sampling time points will be compared using either a one-way analysis of variance with a Dunnett post hoc test for normally distributed data or a Kruskal-Wallis test with a Dunns post hoc test for non-normally distributed data. A $p < 0.05$ will be considered statistically significant.

We will determine in how many participants the measurements of neutrophil phagocytic capacity, plasma cfDNA levels and IG count on day 1 of injury predicted a diagnosis of sepsis. This proportion will be reported along with 95% CIs calculated by binomial exact methods. The accuracy of this day 1 biomarker model will be determined through standard estimates of sensitivity, specificity, predictive values and likelihood ratios, with 95% CIs. We will investigate the value added by this day 1 biomarker test when compared with information obtained from other potentially informative sets of biomarkers (eg, revised Baux score). These combinations of tests will be assessed using logistic regression models. Using multivariable logistic regression analysis, we will also generate predictive probabilities for various combinations of these candidate biomarkers. This logistic modelling will aim to derive a prognostic regression function (ie, probability of sepsis) using the day 1 values of each candidate biomarker and other demographic or burn-related variables. Models of varying complexity may be compared through ROC analyses. Longitudinal data collection will enable us to examine how the performance of these models change over time. Limitations associated with the logistic regression approach lie mainly in its generalisability to other data sets. Thus, techniques such as bootstrapping to enhance generalisability will be applied for model validation. In sensitivity analyses, missing data will be estimated by multiple imputation and maximum-likelihood methods, as appropriate, in order to explore the potential bias and reduced statistical power associated with listwise deletion.

Serial sampling will allow for repeated measures analyses of our candidate biomarkers, enabling us to model their kinetics over time and examine how this differs between septic and non-septic patients. For the analysis of longitudinal data, continuous outcomes will be analysed using linear regression to compare the septic and non-septic groups, with mean differences and 95% CIs reported. For other outcomes, analysis of daily measurements will be performed using generalised linear mixed models to account for the hierarchical structure of the data. Summaries of model output will be provided with model effects presented both as estimates with associated 95% CIs and also, where appropriate, as plots to visualise the time course of fitted outcomes.

## Data storage and retention

Following receipt of consent, patients will be assigned a sequential study number. This non-identifiable number will be used at all times to ensure patient confidentiality. A record of study numbers and patient details will be held at the Queen Elizabeth Hospital Research Office and will only be known to the chief and principal investigators and research personnel. The chief investigator of a particular research site will be responsible for the secure storage of all study related documents (eg, protocols, PIL, general practitioner (GP) letters, consent forms and CRF) in accordance with current International Council on Harmonisation Good Clinical Practice guidelines. All data and study related information will be stored for 15 years after completion of the study. All documents will be archived in accordance with the University Hospitals Birmingham Foundation Trust (UHBFT) archiving procedures.

## Patient and public involvement

Members of a patient and public involvement group were involved in discussions around the design of the study. The group were asked for their opinions on the frequency of patient blood sampling and provided input on the content of the PIL.

## Ethics and dissemination

The SIFTI-2 study received ethical approval from the West Midlands, Coventry and Warwickshire Research Ethics Committee on 7 June 2016 (REC reference: 16/WM/0217) and will be conducted according to the Declaration of Helsinki (2008). Findings from the study will be presented at national and international conferences, and submitted for publication in peer-reviewed clinical and academic journals.

**Author affiliations**
[1] National Institute for Health Research Surgical Reconstruction and Microbiology Research Centre, University of Birmingham, Birmingham, UK
[2] Institute of Inflammation and Ageing, University of Birmingham College of Medical and Dental Sciences, Birmingham, UK
[3] University Hospitals Birmingham NHS Foundation Trust, Birmingham, UK
[4] St Andrews Centre for Plastic Surgery and Burns, Mid and South Essex NHS Foundation Trust, Essex, UK
[5] Birmingham Children's Hospital NHS Foundation Trust, Birmingham, UK
[6] Scar Free Foundation, Birmingham, UK

**Acknowledgements** The authors would like to acknowledge the Queen Elizabeth Hospital Birmingham Charity for funding the purchase of the Sysmex XN-1000 haematology analyser and Mr David Spooner of the patient and public involvement group for his helpful advice on protocol design.

**Contributors** NM is the chief investigator and was responsible for the design and development of the SIFTI-2 study. JL, DB, YW and PH are principal investigators that assisted in protocol development and/or design. JB assisted in protocol design and will perform statistical analysis of the data. JH, KCM, KA-T, KP, TH, RI and AB

commented on the protocol and are involved in the acquisition and/or analysis of data. JH and KCM wrote the manuscript. NM, JL, JB, DB and PH critically revised the manuscript. All authors have viewed and approved the final version of the manuscript.

**Funding** JH and JB are supported by the NIHR Surgical Reconstruction and Microbiology Research Centre; KCM and PH are supported by the Scar Free Foundation; KA-T is funded by a scholarship from the Kuwaiti government. This research received no specific grant from any funding agency in the public, commercial or not-for-profit sectors. Award/Grant number is not applicable.

**Competing interests** None declared.

**Patient and public involvement** Patients and/or the public were involved in the design, or conduct, or reporting, or dissemination plans of this research. Refer to the Methods section for further details.

**Patient consent for publication** Not applicable.

**Provenance and peer review** Not commissioned; externally peer reviewed.

**ORCID iDs**
Jon Hazeldine http://orcid.org/0000-0002-4280-4889
Rizwana Imran http://orcid.org/0000-0002-1641-2478
Janet M Lord http://orcid.org/0000-0003-1030-6786

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
