## [Reviewer comments · BMJ Open]

ARTICLE DETAILS

TITLE (PROVISIONAL)	A Multicentre, Longitudinal, Observational Cohort Study to Examine the Relationship between Neutrophil Function and Sepsis in Adults and Children with Severe Thermal Injuries: a protocol for the Scientific Investigation of the Biological Pathways Following Thermal Injury-2 (SIFTI-2) study.
AUTHORS	Hazeldine, Jon; McGee, Kirsty; Al-Tarrach, Khaled; Hassouna, Tarek; Patel, Krupali; Imran, Rizwana; Bishop, Jon; Bamford, Amy; Barnes, David; Wilson, Yvonne; Harrison, Paul; Lord, Janet; Moiemmen, Naiem

VERSION 1 – REVIEW

REVIEWER	Thakkar , Rajan Nationwide Children's Hospital, Pediatric Surgery
REVIEW RETURNED	05-Jun-2021

GENERAL COMMENTS	I commend the authors for putting together this study protocol and obtaining funding. I am not sure the protocol itself is worth publication. I am enthusiastically looking forward to the results of this study which findings either way will make and warrant publication. Again congrats to the authors and looking forward to the study results.
---

REVIEWER	Tredget, Edward E. University of Alberta Faculty of Medicine & Dentistry, Department of Surgery
REVIEW RETURNED	28-Jun-2021

GENERAL COMMENTS	This manuscript is a description of a study proposal to examine the usefulness of three measures in the blood stream of adult and paediatric burn patients to predict sepsis as defined on the basis of a consensus conference in 2007. The document is a proposal to conduct research and thoroughly outlines the plan but does not contain any data. The topic is important and interesting. The authors have conducted a preliminary study using their approach which is published and becomes the basis for sample size calculation for the follow siftie II study proposal. Concerns with the proposal.  1. It is unclear if this journal will publish research proposals that contain no data or actual patient information. 2. The study is written in the future tense, we will do ... 3. The criteria for sepsis in burns is based on a 2007 consensus of opinions. Since then challenges have been advanced to address the complexities of recognizing the constellation of parameters from the
--

	ABA definition and simpler methods have been proposed with potentially superior discrimination for definition of sepsis. See Rech et al, Burn Care Res 2017 38(5):3112-3116. 4. Similarly, paediatric burn sepsis is different in many ways than adults and the manuscript does not well describe how sepsis will be recognized in this population. The document lacks recognition of the paper by Kraft et al, Ann Surg 2014;259:381-7, where the incidence and outcomes of multi organ failure and sepsis in paediatric burn patients in a large number of patients is reviewed. 4. It is unclear what the role of blood cultures will be in the study, when are they taken, how are they interpreted especially in paediatric burn patients? 5. Similarly, how will burn wound sepsis be diagnosed and its presence influence outcome etc?
--	---

REVIEWER	Toliver-Kinsky, Tracy University of Texas Medical Branch
REVIEW RETURNED	11-Aug-2021

GENERAL COMMENTS	The purpose of the described study is to validate neutrophil phagocytic capacity, immature granulocyte counts, and circulating cell-free DNA as early biomarkers of sepsis in adult and pediatric burn patients. This builds on a previously published study examining the use of these early immune markers to predict development of sepsis in burned adults. Overall, the study design is well-described but there are a few clarifications that would strengthen the protocol. The stated objective is to identify early biomarkers, with a specific focus on day 1 post-burn. The description of data analysis only describes the use of day 1 data. However, samples are being collected daily for the first 14 days post-burn, then at 28 days, and 3, 6, 12, and 24 months. There is no justification for the collection of samples beyond the immediate post-burn period. The authors should provide a justification for longitudinal sample collection and a description of how these data will be analyzed and how they relate to the study objective of identifying early biomarkers of sepsis. The study will recruit 245 burn patients. Sample size for this population is justified based on prior studies. It is assumed that 245 burned adults and 245 burned children (not 245 total) will be recruited but that should be stated. Additionally, samples will be collected from healthy controls, including 100 adults and 10 children. There is no rationale provided for the healthy control sample sizes. Given the developmental differences between children of age 1 vs 16, it is surprising that only 10 healthy children will be recruited. Sample sizes in the control groups should be justified. There was no mention of age-matching between control and injured groups. Given the wide age spans (1-16 for pediatric and 17-99 for adults), and the increased susceptibilities of very young and aged patients to sepsis, the investigators should consider age-matching between study groups. The anticipated diversity in gender and race or ethnicity of their recruited subjects is not described. The dates of the study and estimated time frame for patient enrollment are not provided. Regarding the measurements, it is assumed that the IG counts will
---

	be run right away, and instrument quality control is described. However, for the cell free DNA and PCR measurements, information on sample storage and analysis should be provided; i.e.-how long will plasma and samples/RNA be stored prior to running the assays? More details on standardization and quality control should be provided. For cfDNA and PCR measurements, will cumulated samples be run together? If so, how frequently? Will samples from healthy controls always be included for a reference? What controls and protocols are in place to standardize measurements over time? For phagocytosis measurements, more experimental details should be provided. For example, how will neutrophils be distinguished from other phagocytes? It is assumed that FSC/SSC, and not cell type-specific antigen staining, will be used but this should be stated. Since gating can be somewhat subjective, how will this be standardized over time and different users? Will appropriate controls be included to distinguish internal (phagocytosed) vs externally bound bacterial particles? What standardization, normalization and quality control measurements will be used to ensure comparable measurements over time, since samples will likely be run individually as they become available, and healthy controls may not always be available to run simultaneously? This is important as FACS gating parameters can be somewhat subjective. This is less of a concern for the percent positive calculations but mean fluorescence intensity, which is included in the phagocytic capacity calculation, is dependent on the average fluorescence intensity per bacterial particle, and there is variability in bacterial labelling efficiency and fluor intensity between labelled batches. How will this be controlled and normalized over time? Effects of injury on neutrophil function, release of immature granulocytes, and cell-free DNA have been previously reported and typically correlate with severity of injury, which correlates with risk for subsequent sepsis. Based on their previously reported study using a smaller sample size, it is not clear how much additional predictive value these immune measures would offer compared to the rBAUX score (based on age, burn size and depth and presence of inhalation injury). While the combined use of these immune markers with rBAUX will also be explored, the actual impact of these additional measures on clinical care may be incremental; this should be acknowledged as a limitation. Although the study only requires low-risk blood draws and was approved by a research ethics committee, there are some concerns with the consent process. Specifically, children may consent themselves if they meet the Gillick Competence criteria, which appear to involve subjective assessments of maturity and understanding. Since very young burn victims are included in this study (age range 1-16), can the authors provide more details and justification for self-consent by children? Additionally, the use of "nominated consultees," medical professionals that are not involved in the study, to consent on behalf of an adult that lacks mental capacity and a personal/familiar consultee, is concerning. The authors should also elaborate on the rationale for this consent process.
--	--

VERSION 1 – AUTHOR RESPONSE

Reviewer: 1
Dr. Rajan Thakkar, Nationwide Children's Hospital

Comments to the Author:

I commend the authors for putting together this study protocol and obtaining funding. I am not sure the protocol itself is worth publication. I am enthusiastically looking forward to the results of this study which findings either way will make and warrant publication. Again congrats to the authors and looking forward to the study results.

We thank the reviewer for their comments. We have been informed through the editor that BMJ Open does publish study protocols for all study designs.

Reviewer: 2
Dr. Edward E. Tredget, University of Toronto

Concerns with the proposal.

- 1. It is unclear if this journal will publish research proposals that contain no data or actual patient information.**

It has been confirmed to us by the editor that BMJ Open will publish such articles.

- 2. The study is written in the future tense, we will do ...**

As instructed in correspondence received from the editor, we have not changed to using past tense in the revised manuscript.

- 3. The criteria for sepsis in burns is based on a 2007 consensus of opinions. Since then challenges have been advanced to address the complexities of recognizing the constellation of parameters from the ABA definition and simpler methods have been proposed with potentially superior discrimination for definition of sepsis. See Rech et al, Burn Care Res 2017 38(5):3112-316.**

The authors would like to thank the reviewer for highlighting concerns relating to the use of the 2007 ABA consensus to define sepsis, and thank them for bringing to our attention the work of Rech et al, who performed a retrospective cohort study to develop an algorithm for the identification of sepsis and sepsis with organ dysfunction/septic shock in burn-injured patients.

The primary objective of the SIFTI-2 study is to prospectively validate the findings reported in our previous observational based cohort study that identified neutrophil phagocytic activity, IG count and plasma cfDNA levels as potential early biomarkers of post-burn sepsis (*Hampson et al; PMID:27232244*). In our previous study, we defined the development of this secondary complication using the 2007 ABA consensus. Thus, in an effort to validate our original observations, we feel that it is necessary to use the same definitions and approaches to diagnose post-burn sepsis in this follow-up study.

However, it should be noted that in addition to recording data that will allow for the diagnosis of sepsis using the 2007 ABA consensus, we will also be recording other clinical scores for our burns patients such as the Early Warning Score (EWS), Multiple organ dysfunction syndrome (MODS), SOFA score, Marshall Score and SIRS Score. All of these parameters could be used in future analyses should a reliable and validated burns sepsis score be available in the future. A statement

that these additional scores will be recorded for patients enrolled into the SIFTI-2 study has been added into the revised manuscript (Page 10, Lines 420-426).

The algorithm developed by Rech et al was validated in a single centre study that used the 2007 consensus and SOFA score as a "reference standard". In the absence of a reliable and validated sepsis score, such as Sepsis-3, for burns patients, we believe that it is still reasonable to use the ABA 2007 consensus for the reasons we have outlined above.

- 4. Similarly, paediatric burn sepsis is different in many ways than adults and the manuscript does not well describe how sepsis will be recognized in this population. The document lacks recognition of the paper by Kraft et al, Ann Surg 2014;259:381-7 PubMed, where the incidence and outcomes of multi organ failure and sepsis in paediatric burn patients in a large number of patients is reviewed.**

The authors recognise the difficulty of applying the same sepsis scoring for adults and children. In an effort to combat this issue, a daily assessment of organ dysfunction, alongside an ABA 2007 consensus score, will be recorded as described by Kraft et al. A statement outlining this point has been added into the revised manuscript (Page 10, Lines 420-426).

- 5. It is unclear what the role of blood cultures will be in the study, when are they taken, how are they interpreted especially in paediatric burn patients?**

We thank the reviewer for this comment and apologise for the lack of clarity regarding this issue in our original submission. Blood cultures will be taken whenever the temperature of a patient is $>38.5^{\circ}\text{C}$ and they will be acquired from a clean venepuncture site where possible using a sterile technique. If such a site is not accessible, due to the patients injuries, then blood cultures will be obtained from a pre-existing central venous catheter. The results will be interpreted following detailed discussions with the microbiology team. A sentence outlining this approach has been included in the revised manuscript (Page 9, Lines 411-414).

- 6. Similarly, how will burn wound sepsis be diagnosed and its presence influence outcome etc?**

Burn wound infection/sepsis will be documented in the case report form (CRF) after assessment by a senior surgeon who will consult information attained from the relevant microbiology wound swabs/tissue biopsies, together with the sepsis definition described above. We will also record the time to 95% wound healing as an overall measure of the patients care but also wound related complications such as wound sepsis. Both of these points have been added into the revised submission (Page 9, Lines 415-418).

It is unpredictable how the presence of sepsis may influence outcome in any individual patient; sometimes a standard course of antibiotics will produce a good clinical response with only minor effect on outcome, such as a slightly longer ICU stay. In other cases, sepsis can have major effects on a patients outcome including graft loss, ARDS, multi-organ dysfunction and potentially death. We hope that our study may yield results that will help to answer this question, along with addressing why the physiological response to sepsis varies between different patients.

Reviewer: 3
Dr. Tracy Toliver-Kinsky, University of Texas Medical Branch

- 1. The stated objective is to identify early biomarkers, with a specific focus on day 1 post-burn. The description of data analysis only describes the use of day 1 data. However, samples are being collected daily for the first 14 days post-burn, then at 28 days, and 3, 6, 12, and 24 months. There is no justification for the collection of samples beyond the immediate post-burn period. The authors should provide a justification for longitudinal sample collection and a description of how these data will be analysed and how they relate to the study objective of identifying early biomarkers of sepsis.**

We thank the reviewer for this comment and recognise the lack of detail relating to the reasons behind the longitudinal aspect of the study in our original submission. Across paediatric and adult burns, a significant number of studies have undertaken serial blood sampling to investigate the short (<1-7days), medium (8-34 days) and long (35-1100 days)-term inflammatory response to thermal injury and its relationship with patient outcome (e.g. Bergquist et al. 2019, *Dehne et al. 2002, Finnerty et al. 2008, Hur et al. 2015, Jeschke et al 2008, Jeschke et al 2011, Vindenes et al. 2011*). In contrast, there is very limited data relating to the acute and long-term effects of thermal injury on the immune system (e.g. *Hampson et al. 2017, Johnson et al. 2020, Mulder et al 2020*). Thus, generating data on the kinetics of the immune response to burn injury, via serial sampling in both our adult and paediatric cohorts, will allow us to not only establish how long burn-induced changes in immunity persist for (when compared to healthy subjects), but to also compare the immune response of septic and non-septic burns patients over time. Generating and interrogating such novel data may provide mechanistic insights into the long-term health complications reported by survivors of burn trauma such as their increased susceptibility to respiratory infections (e.g. *Fear et al. 2017*). A summary of this response has been added into the revised manuscript (Page 7, Lines 303-311).

In the context of identifying early biomarkers of sepsis, the collection of longitudinal data will allow us to test whether the power of our candidate biomarkers, in respect of their ability to discriminate between septic and non-septic patients, changes/wanes of time. Based on our previous study, we speculate that data obtained from samples acquired within 24 hours of injury will provide the greatest discriminatory power, which if correct, will demonstrate the importance of undertaking “early” blood sampling when attempting to identify those patients at risk of post-injury complications. Daily blood sampling will also tell us at what time-points post-burn, our candidate biomarkers return to “normal” levels (i.e. comparable to those measured in healthy volunteers) and how they are affected by surgical procedures and in the days preceding and following septic episodes.

For the analysis of longitudinal data, continuous outcomes will be analysed using linear regression to compare the septic and non-septic groups, with mean differences and 95% confidence intervals reported. For other outcomes, analysis of daily measurements will be performed using generalised linear mixed-models to account for the hierarchical structure of the data. Summaries of model output will be provided with model effects presented both as estimates with associated 95% confidence intervals and also, where appropriate, as plots to visualise the time course of fitted outcomes. This description of data analysis has been included in the revised submission (Page 10, Lines 455-462).

- 2. The study will recruit 245 burn patients. Sample size for this population is justified based on prior studies. It is assumed that 245 burned adults and 245 burned children (not 245 total) will be recruited but that should be stated. Additionally, samples will be collected from healthy controls, including 100 adults and 10 children. There is no rationale provided for the healthy control sample sizes. Given the developmental differences between children of age 1 vs 16, it is surprising that only 10 healthy children will be recruited. Sample sizes in the control groups should be justified.**

We thank the reviewer for highlighting the confusion relating to the patient sample size in our original submission. To be clear, the total sample size of the patient cohort for the SIFTI-2 study will be 245, which will include both adult and paediatric age groups. It would not be possible within the UK to

recruit 245 severely burned children, even with a multicentre study, due to the thankfully small numbers of these serious injuries in children in this country. We have clarified that the total sample size of the SIFTI-2 study is 245 in the revised manuscript (Page 7, Lines 295-297).

We understand your concerns about the difference in the sizes of the two groups of healthy volunteers. Adult healthy volunteers can be recruited from age 16 years, with no upper age limit. This allows for diverse sampling of the adult population, with all the variations that this will encompass, including general physical condition, concurrent medical conditions and drug therapies, past exposure to disease or injury and different degrees of physiological ageing. This is why a large group, of up to 100 subjects, will be recruited. These reasons have now been included in the revised manuscript (Page 5, Lines 201-204)

In the paediatric age group, the healthy volunteers will represent a more homogeneous group in terms of age range and lack of other health conditions or ongoing medical treatments. It would therefore be anticipated that the results of our experimental assays for the paediatric group will demonstrate much less variability than those from the adult group. In addition, it should be noted that recruiting healthy children is quite a complex process. Given that the number paediatric burns patients recruited into the study will be significantly less than our adult cohort, ten paediatric healthy controls is our best estimate of the numbers we will be able to recruit. A summary of this point has been included in the revised submission (Pages 5-6, Lines 213-217)

- 3. There was no mention of age-matching between control and injured groups. Given the wide age spans (1-16 for paediatric and 17-99 for adults), and the increased susceptibilities of very young and aged patients to sepsis, the investigators should consider age-matching between study groups.**

We thank the reviewer for highlighting this omission from the original manuscript. We agree fully with this comment and can confirm that both the paediatric and adult healthy controls that will be recruited into the SIFTI-2 study will be aged-matched to the relevant study group. The enrolment of age-matched healthy volunteers has now been stated in the revised submission (Page 5, Lines 197-198).

- 4. The anticipated diversity in gender and race or ethnicity of their recruited subjects is not described.**

Screening of the enrolment log generated for patients recruited into our original observational study (*Hampson et al; PMID:27232244*) revealed a predominantly male (61%) and Caucasian (79%) patient cohort. Given that the West Midlands Regional Burns Centre will again serve as the primary site of patient recruitment, we anticipate a similar patient demographic for the SIFTI-2 study. This statement has now been included in the revised manuscript (Page 5, Lines 173-176).

- 5. The dates of the study and estimated time frame for patient enrolment are not provided.**

We thank the reviewer for drawing our attention to this omission. The first patient was enrolled into the SIFTI-2 study on the 27th November 2016. The estimated date for completion of the study is December 2025. This information has now been included in the revised manuscript (Page 7, Lines 300-301).

- 6. Regarding the measurements, it is assumed that the IG counts will be run right away, and instrument quality control is described. However, for the cell free DNA and PCR measurements, information on sample storage and analysis should be**

provided; i.e.-how long will plasma and samples/RNA be stored prior to running the assays? More details on standardization and quality control should be provided. For cfDNA and PCR measurements, will cumulated samples be run together? If so, how frequently? Will samples from healthy controls always be included for a reference? What controls and protocols are in place to standardize measurements over time?

We thank the reviewer for these comments. We will determine the concentration of cell free DNA (cfDNA), mitochondrial-derived DNA (mtDNA) and nuclear-derived DNA (nDNA) in platelet free plasma (PFP) that will be prepared from citrate anti-coagulated blood samples. Once generated, PFP will be stored at -80°C prior to analysis.

CfDNA concentrations will be determined in duplicate PFP samples using an in-house SYTOX® Green dye based fluorometric assay. Included within each assay, which will be performed in 2 phases (phase 1: analysis of samples collected 0-28 days post-burn; phase 2: analysis of samples collected at months 3-24 post-burn) will be: (1) a λ -DNA standard curve (0-1000 ng/ml) for calibration, (2) PFP samples from healthy volunteers to serve as control samples and (3) a “buffer only” negative control. The mean fluorescent value derived from the negative control will be subtracted from all HC and patient data prior to the calculation of DNA concentration. PFP samples from a selection of healthy volunteers and patients will be analysed across our cfDNA assays to allow for the calculation of inter-assay and intra-assay coefficients of variation, which based on current data stand at 5.3% and 5.1% respectively. To test whether our in-house assay for the quantification of cfDNA was comparable to that of a CE marked fluorometric assay, we have previously measured cfDNA levels in PFP samples using both our in-house assay and a commercially available CE marked assay provided by Trillium Diagnostics. Analysis of 32 PFP samples revealed a strong positive correlation between the concentrations of cfDNA measured using these two assays (Spearman R = 0.78, P<0.0001).

Isolated using the commercially available QIAamp DNA mini kit, DNA extracted from PFP samples will be subjected to RT-PCR to allow for the determination of mtDNA and nDNA levels. Included within each assay, which will be performed in 2 phases (phase 1: analysis of samples collected 0-28 days post-burn; phase 2: analysis of samples collected at months 3-24 post-burn) will be: (1) standard curves generated from DNA isolated from purified mitochondria and nuclei to allow for the quantification of mtDNA and nDNA concentrations, (2) DNA obtained from PFP samples of healthy volunteers to serve as control samples and (3) a “buffer only” negative control to test for the presence of primer dimers. PFP samples from healthy volunteers and patients will be analysed across assays in order to allow for the calculation of inter-assay and intra-assay coefficients of variation. The primers sets that will be used to amplify mtDNA and nDNA are specific for the genes encoding cytochrome b and β -globin respectively. These primer sequences have no significant homology with DNA found in any bacterial species according to information provided by the Basic Local Alignment Search Tool (BLAST).

A summary of all the above points has been added to the revised submission (Pages 8-9, Lines 362-382).

- 7. For phagocytosis measurements, more experimental details should be provided. For example, how will neutrophils be distinguished from other phagocytes? It is assumed that FSC/SSC, and not cell type-specific antigen staining, will be used but this should be stated. Since gating can be somewhat subjective, how will this be standardized over time and different users? Will appropriate controls be included to distinguish internal (phagocytosed) vs externally bound bacterial particles? What standardization, normalization and quality control measurements will be used to ensure comparable measurements over time, since samples will likely be run individually as they become available, and healthy controls may not always be available to run simultaneously? This is**

important as FACS gating parameters can be somewhat subjective. This is less of a concern for the percent positive calculations but mean fluorescence intensity, which is included in the phagocytic capacity calculation, is dependent on the average fluorescence intensity per bacterial particle, and there is variability in bacterial labelling efficiency and fluor intensity between labelled batches. How will this be controlled and normalized over time?

We thank the reviewer for highlighting these omissions from our original submission. The reviewer is correct in that, as stated in manufacturer's instructions, neutrophils will be identified according to their distinctive forward scatter/sideward scatter properties when using the PhagoTest kit supplied by BD Biosciences.

For the acquisition of raw data, a flow cytometry template (containing gates, event limits etc) has been generated and is used by all investigators that are measuring neutrophil phagocytosis in patient and healthy volunteer blood samples. The experimental protocol we use to assess neutrophil phagocytic activity includes a step where trypan blue is added to blood samples following their stimulation with bacteria. Trypan blue is added to "quench" fluorescent signals that would otherwise be emitted by bacteria externally bound to the membranes of neutrophils. This approach ensures that the fluorescent signal we detect on the flow cytometer is derived only from phagocytosed bacteria, and does not reflect an overestimation of phagocytic activity that would otherwise arise from the combined fluorescence of external and phagocytosed bacteria.

Prior to the analysis of every patient and healthy volunteer blood sample, BD™ CS&T beads (BD Biosciences) are run on our flow cytometer. These beads allow for a standardised method of performing quality control in relation to the optics, electronics and fluidics of our instrument. Variability in labelling efficiency/fluorescence intensity of bacteria between batches will be checked by analysing the phagocytic activity of the same patient/healthy volunteer blood sample against existing and new kits with different lot/batch numbers.

A summary of the above points have now been included in the revised manuscript (Page 8, Lines 343-359).

- 8. Effects of injury on neutrophil function, release of immature granulocytes, and cell-free DNA have been previously reported and typically correlate with severity of injury, which correlates with risk for subsequent sepsis. Based on their previously reported study using a smaller sample size, it is not clear how much additional predictive value these immune measures would offer compared to the rBAUX score (based on age, burn size and depth and presence of inhalation injury). While the combined use of these immune markers with rBAUX will also be explored, the actual impact of these additional measures on clinical care may be incremental; this should be acknowledged as a limitation.**

We thank the reviewer for raising this point. From a statistical perspective, we are in agreement that the additional predictive value may be fairly small, but we do not know if that will be the case until we conduct the analysis. In our previous study, the 'preferred' model was based on 24 sepsis cases in 33 patients and had an AUROC 95% CI from 0.854 to 1 and, due to the small sample size, likely suffered from over-fitting and an inflated AUROC. By increasing our sample size to 245 participants, we now expect to see around 132 cases of sepsis (assuming a 10% drop-out rate and sepsis prevalence of 60%) and therefore should not suffer from over-fitting. By increasing our sample size by more than 5x that of the original study, we are now powered to reliably rule out a sensitivity less than 84% and a specificity less than 59% if the 'true' sensitivity is at least 90% and the 'true' specificity is at least 70% (assuming a prevalence of 60%).

In our original study, we found that models built on combinations of our candidate biomarkers and a patients rBAUX score provided greater discriminatory power to distinguish between our septic and non-septic cohorts than any of the measurements alone. However, we appreciate that what we saw in our first study could have been a chance finding due to the small sample size. By undertaking this

follow-up study, with a greatly increased sample size, we are hoping to confirm that the relationship between our candidate biomarkers and the risk for subsequent sepsis, still holds and to estimate the prognostic performance of this model with greatly improved accuracy compared to our initial study. Equally, this study may turn out to show that the relationship is not as strong as we thought and that there may be some other, confounding, mechanism at play. We are of the opinion that there is still enough uncertainty about the true utility of neutrophil phagocytic activity, immature granulocyte number and cell-free DNA as prognostic indicators for the risk of sepsis to justify the SIFTI-2 study and obtain much more precise estimates of the relationship between these biomarkers and sepsis.

- 9. Although the study only requires low-risk blood draws and was approved by a research ethics committee, there are some concerns with the consent process. Specifically, children may consent themselves if they meet the Gillick Competence criteria, which appear to involve subjective assessments of maturity and understanding. Since very young burn victims are included in this study (age range 1-16), can the authors provide more details and justification for self-consent by children? Additionally, the use of "nominated consultees," medical professionals that are not involved in the study, to consent on behalf of an adult that lacks mental capacity and a personal/familiar consultee, is concerning. The authors should also elaborate on the rationale for this consent process.**

We apologise to the reviewer for the lack of detail relating to the consenting procedure for paediatric burns patients in our original submission. All children will be consented in the SIFTI-2 study by their parent or legal guardian. If the child is deemed as being Gillick competent, they will also consent for themselves. This is to ensure that children of a certain level of maturity understand what is being proposed and agree with joining the study. If any child, who was deemed Gillick competent, does not wish to give consent, then they would not be recruited into the study, even if the legally responsible adult had consented. Thus, no child would self-consent without adult consent. The child's consent is a matter of confirming their agreement with the adult consent. If the child is not consenting, they do not join the study. In the UK, Gillick competent children will often consent for procedures, such as surgery, alongside and in conjunction with the responsible adult. A summary of these points have been added into the revised submission (Page 6, Lines 255-264).

Patients with major burns are usually admitted to a burns centre that is located some distance away from the site at which their injuries were sustained. They often arrive at night, sometimes by air and no accompanying family or next of kin are available at that time. Most of the patients are unconscious (sedated and ventilated) either on, or shortly after, their arrival at the emergency department. In this respect, they lack mental capacity to take part in a consent process. Due to the circumstances of the injury, it is common to have some difficulty in tracing next of kin or family members. All efforts are made to contact family as soon as possible, despite a lack of information being available. When family members are aware of the injury, they will rapidly be in touch with the hospital. They would then be involved in the consent process. In other circumstances, next of kin will usually have been identified within 48 hours. However, this is not always the case, depending on the patients individual circumstances.

The rationale for the use of "nominated consultees" is that the systemic physiological response to burn trauma is ongoing from the time of injury. As blood samples obtained close to the time of injury provide useful data that may relate to patient outcomes (e.g. *Hampson et al; PMID:27232244*), this process allows for earlier recruitment whilst efforts are being made to contact their next of kin. As soon as family members are contacted, our research nurses speak with them with regards to patient recruitment. If they are in agreement, then sample collection continues according to study protocol. If they are not in agreement, then the patient would be withdrawn from the study. Using this system of consent allows data to be obtained from samples collected during the early "window" directly after injury that would otherwise be missed. A summary of the benefits of this approach are now included in the revised protocol (Page 6, Lines 229-241).

VERSION 2 – REVIEW

REVIEWER	Tredget, Edward E. University of Alberta Faculty of Medicine & Dentistry, Department of Surgery
REVIEW RETURNED	13-Sep-2021

GENERAL COMMENTS	The authors have addressed many concerns in the revised manuscript satisfactorily; however, issues remaining are the following: 1. It is unclear how samples will be batched, cryoprotected for storage and the effect of such handling on the stability of the analysis given the long period of duration of the study.2. Also, it is noted that the study has begun in 2016 and since the protocol has been revised. How valid are samples obtained and cases enrolled prior to the revision and acceptance of the final protocol?
--

REVIEWER	Toliver-Kinsky, Tracy University of Texas Medical Branch
REVIEW RETURNED	13-Sep-2021

GENERAL COMMENTS	The authors have provided additional protocol details that clarified their goals, study design and methods and thus have addressed the previous concerns regarding patient recruitment and consent and assay quality control over time. There are no major concerns. Completion of this study will provide new information on the long-term effects of burn injury on granulocyte function, and the relationship between early granulocyte dysfunction and susceptibility to sepsis.
---

VERSION 2 – AUTHOR RESPONSE

Reviewer: 2

Dr. Edward E. Tredget, University of Alberta Faculty of Medicine & Dentistry

1. It is unclear how samples will be batched, cryoprotected for storage and the effect of such handling on the stability of the analysis given the long period of duration of the study.

We thank the reviewer for this comment. The primary objective of our study is to validate neutrophil phagocytic activity, IG count and plasma cfDNA levels as early biomarkers of sepsis in adult and paediatric thermally-injured patients. Data relating to two of these candidate biomarkers (IG count and neutrophil phagocytic activity) are obtained on the day of patient sampling, with blood samples processed immediately upon their arrival into the laboratory. Thus, by performing these analyses in real-time, we bypass the need for cryopreservation and storage, thereby ensuring that our assessment of immune function post-burn is not affected by these potential variables. A description of the time-frame of our analyses of immune function has now been included in the revised submission (Page 8, Lines 336-339).

Concentrations of cfDNA will be measured in platelet free plasma (PFP) samples that are stored at -80°C immediately following their preparation from whole blood on the day of patient sampling. Assays to measure cfDNA are performed in 2 batches: (1) samples collected from day 1 to day 28 and (2) samples collected from month 3 to month 24. In terms of long-term storage for samples that will be analysed in batch 2, results from our previous observational study (Hampson et al, PMID: 27232244),

in which cfDNA levels were measured in PFP samples collected from month 2 to month 12 post-injury, revealed that our preparation and storage protocols allow for the measurement and isolation of pure high quality DNA that can be analysed by fluorometry and used in PCR assays. The inclusion in our assays of PFP samples obtained from HCs that will be stored under the same conditions as our patient samples will assist in ensuring the validity of our data when comparisons are made between our burns and HC cohorts. A description of the time-frame of our analyses can be found in the section that discusses our measurement of cfDNA in PFP samples (Page 9, Lines 368-371).

2. Also, it is noted that the study has begun in 2016 and since the protocol has been revised. How valid are samples obtained and cases enrolled prior to the revision and acceptance of the final protocol?

We thank the reviewer for this comment. Revisions made to the protocol were in relation to allowing for the long-term assessment of scarring in patients post-burn. Thus, the amendments did not affect any of the laboratory based analyses that are performed within the study. For this reason, the data collected in the early part of the study remains comparable to the data that is currently being generated and data that will be acquired from future patients enrolled into the SIFTI-2 study.